# Tracking HPV Infection, Associated Cancer Development, and Recent Treatment Efforts—A Comprehensive Review

**DOI:** 10.3390/vaccines11010102

**Published:** 2023-01-01

**Authors:** Shiza Malik, Ranjit Sah, Khalid Muhammad, Yasir Waheed

**Affiliations:** 1Bridging Health Foundation, Rawalpindi 46000, Pakistan; 2Department of Microbiology, Tribhuvan University Teaching Hospital, Institute of Medicine, Kathmandu 44600, Nepal; 3Department of Biology, College of Sciences, United Arab Emirates University, Al Ain 15551, United Arab Emirates; 4Office of Research, Innovation, and Commercialization (ORIC), Shaheed Zulfiqar Ali Bhutto Medical University, Islamabad 44000, Pakistan; 5Gilbert and Rose-Marie Chagoury School of Medicine, Lebanese American University, Byblos 1401, Lebanon

**Keywords:** human papillomavirus (HPV), co-infections, carcinogenesis, host-interaction, immune response, vaccines, drugs

## Abstract

Human papillomaviruses (HPVs) are high-risk causative factors for HPV infection. This infection does not come alone; it is often seen with co-infection with other viruses and acts as a causative agent for several malignancies. The major purpose of this comprehensive study was to highlight some recent advances in biotechnology associated with HPV infection, including understanding its host interactions and cancerous progression. A systematic research strategy was used to gather data from recent, and the most advanced published electronic sources. The compiled data explain the recent understanding of biology, host–viral interaction cycles, co-infection with other viral diseases, and cellular transformation toward malignancies associated with HPV. In recent years, some vaccination protocols have been introduced in the form of live attenuated, subunit, and DNA-based vaccines. Moreover, some strategies of nanotechnology are being employed to synthesize drugs and vaccines with a whole new approach of plant-based products. The data are immense for the proposed research question, yet the need is to implement modern follow-up screening and modern therapeutics at the clinical level and to conduct wide-scale public awareness to lessen the HPV-related disease burden.

## 1. Introduction

Human papillomavirus (HPV) is one of the most potent viruses that cause sexually transmitted diseases in a large-scale population worldwide. Approximately 120 subtypes of HPV have been identified through various research studies, of which ~40 lead to infectious cycles in the host [1]. Some subtypes are divided into low- and high-risk HPV, depending upon the sort of disease or cancer they inflict on the host. Young adults are especially vulnerable to multiple strains associated with critical infection, while the risk of infection decreases with age and with the development of immunity against the virus [2].

The clinical manifestations underlying HPV infection range from benign lesions caused by low-risk oncogenic HPV (LR-HPV) to critical cancers such as head and neck, oropharyngeal, penile, vulvovaginal, anal, and, most importantly, cervical cancers caused by high-risk oncogenic HPV (HR-HPV) [3]. High-risk oncogenic HPV categories are the major reason behind approximately 5% of human cancer incidences [4]. Statistical data gathered over recent years help to infer that HPV-associated infections and cancers are a threat to human life and a burden on healthcare. According to an estimate provided by the CDC, on average, ~47,200 new cases of cancers annually are identified, of which ~80% are associated with HPV infection [5]. However, it is important to note that data tracing occurs mostly in only well-developed countries where proper databases are generated for record-keeping and research purposes. However, the situation is different for developing countries. A considerable gap lacks in statistical approximation for low- and middle-income countries where the virus brings major threats, owing to poor health conditions. Thus, it can easily be deduced that, if proper data tracing is carried out all over the world, the statistical readings might show much more disease than the presently known numbers. The known statistical values indicate the sort of healthcare burden HPV brings to the human population worldwide [6].

In this review, we discuss the biological nature of the HPV virus, its history, host interaction strategy, cellular transformation underlying viral infection, and cancer progression in patients. Moreover, some highlights are also discussed about co-infection with other viruses such as HIV, EBV, and HSV, and with bacteria, along with insights into cancerous development. Some aspects regarding vaccination and therapeutic development, along with possible treatment options for HPV-associated malignancies, are also discussed briefly. The objective is to make the scientific community aware of the latest features of HPV and the new therapeutic options coming to the fore in the healthcare industry.

## 2. Materials and Methods

This review article was generated by systematic research conducted on electronic sources such as Google Scholar, PubMed, NIH (National Library of Medicine), Scopus (Elsevier), Web of Knowledge (Thomson Reuters), Clinical Trial Register, Cochrane Central, and Web of Science. The data from these platforms were acquired from various publications such as original research articles, reviews, short commentaries, book sections, case reports, and letters to the editors. Additionally, statistical data were derived from the official website of the CDC, WHO, and FDA. Finally, the search strategy was designed to incorporate data from 2010 to 2022 and to incorporate the most recent advances related to HPV infections worldwide. A total of 104 papers are included in this review article.

## 3. Results and Discussion

### 3.1. Biology of Human Papillomavirus (HPV)

Human Papillomaviruses (HPVs) are small, non-enveloped viruses, with a double-stranded circular DNA of 8 kb in size. The genome encodes a total of 8 proteins: 2 (L1 and L2) translate for the structural capsid proteins while the remaining 6 (E1, E2, E4, E5, E6, and E7) translate for the non-structural viral regulatory proteins [6]. HPV genomic transcripts also contain non-coding proteins, which play a crucial role in virus–host interaction and infection development via protein–protein interactions [6].

In most cases, HPV causes a transient form of infection that allows slow healing in infected individuals, but in a limited number of cases, serious and long-term infection leads to the most critical health-related conditions [7]. Low-risk HPV leads to minor benign hyperproliferative lesions, while high-risk HPV mostly leads to malignant lesions. Generically, HPV is classified into five genera and is specified by Greek letters—alpha, beta, gamma, mu, and nu—that specify the nucleotide sequences in their genome along with phylogenetic and pathological features [8]. Genus alpha papillomavirus is the major causative agent of cervical, genital, extragenital, and oral cancers [5]. Among some high-risk clades of alpha, genus α-9 is the most critical, among which HPV-16 acts as the major cause of critical viral infection [9]. Apart from sexual behavior, some other factors such as reproductive cycles, heredity, immunity, nutritional status, smoking, and drinking habits may also contribute to the seriousness of viral pathogenesis and malignancies [1].

### 3.2. HPV and Host–Virus Interactions

Multiple in vivo, in vitro, and computational studies have been conducted in past years to understand the biology and nature of host and immune system reactions toward HPV viruses. Such investigations are important to comprehend the biological nature of viruses and to develop an understanding of their genomic alternations that leads to infection development and resistance to host immunity. These insights are important for the development of specific therapeutics and vaccines against viral clinical manifestations. A brief understanding of virus–host interaction and its life cycle is explained below.

HPV–host interaction initiates with viral entry into the undifferentiated cells in the squamous epithelium of the ectocervix. The process mostly occurs as a result of a high-grade mechanical stress such as heat, radiation, or chemical shock that the body may experience [8]. Mechanical stress may be experienced in the form of an injury to the tissues or bones or extended overloading. Chemical shock may be experienced in the form of an internal disbalanced hormonal and enzymatic profile due to disease state (inflammation or infections) or may be due to external intake of drugs or medicines that affect normal organ regulatory systems in the human body. Moreover, heat and radiation shocks, such as ionizing irradiation or UV exposure, may create oxidative stress, responsible for HPV entry and infection of the host cell [10]. The affected cells that undergo shock absorption and viral entry retain the ability to multiply and differentiate. This property is utilized by viral particles to replicate their genome within the host cells. The whole cycle of the virus can be divided into 3 phases, namely productive-, maintenance-, and differentiation-dependent phases [11]. Most critical infections last for 12–18 months and mostly clear spontaneously upon sufficient responses from the immune system of the host, while in other cases, weak immunity leads to insufficient viral elimination and results in persistent infection viral buildup that leads to malignancy [12].

The cycle initiates with the entry of the virus into stem cells of the basal epithelium. Viral internalization is assisted by a complex network of interaction between viral enveloped capsid proteins and the receptor cellular proteins on the host cell surface. This internalization is followed by ligand-activated, complex endocytic pathways inside the host cell [13]. The detailed discussion of this complex mechanism is out of the scope of this review article; thus, we only briefly discuss the process here. Upon entry, the virus utilizes the E1 and E2 genes for maintaining a low number of genomic copies. The viruses replicate its DNA with the help of some proteins driven from the host cell. In addition, the expression of genes E1, E2, E5, E6, and E7 via viral replication helps to maintain the viral genome and induce further proliferation that causes HPV infection in the cellular epithelium [11]. Resultantly, a large number of infectious virions are created that start to circulate the epithelium. The E4 gene is also expressed, which promotes the replication phenomena [13]. The late genetic expression of viral capsid proteins L1 and L2 gathers to assemble capsid proteins for the new viral copies to be released from the infected epithelium layer. This phase is named a productive phase of the virus life cycle [14,15].

In the second phase of maintenance, HPV virions infect cells in epithelial layers and become exposed to the extracellular spaces through microlesions. Viral binding to the basal layer activates keratinocytes in the wound. Here, the virus replicates through the host-DNA machinery to create a low number of copies of virions [15]. These virus particles are then exposed to the immune system to generate a protective response against low viremia before further viral replication and persistence [16]. This phase leads to the maintenance of viruses in the host body. At this stage, expressions of E6, E7, E1, and E2 contribute to episomal maintenance and DNA replication competencies [15].

The virus genome replicates along with host cell replication. During this process, the virus incorporates itself into the rolling-circle mode of DNA replication to amplify thousands of viral copies in the host cell epithelium [17]. Viral proteins also promote de-regulation in normal cell cycles of the host and, thus, subvert the regulatory pathways and modify the cellular signaling environment to facilitate further viral replication [12]. Unlike the maintenance phase, the virus enters the differentiation-dependent phase in an exciting replication mode, with excessive viral protein translation [17]. This delayed viral protein expression in differentiated cells is the major reason for the delay in antigenic expression in host immune cells [18]. This factor plays a major role in delayed immune responses against viremia and the long-term infectious strategy imposed by HPV [19].

E7 plays a critical role during replication competencies in differentiated cells. During this phase, virus copies amplify thousands of copies in a single cell of the host. This replication strategy is the major force behind the escape mechanism of the virus against the host’s humoral and cellular responses [20]. E1, E4, and E5 proteins contribute to late viral functions in the differentiation stages. E6 and E7 proteins play major roles in complex cellular protein formation that control cell cycle function and incur genomic stability to viral particles [20]. Capsid proteins L1 and L2 are also formed at this stage and help to release viral assemblies and complete virus shedding through the epithelium layers into the cellular environment with the least lysis or necrosis rates [20,21].

The aforementioned stages of replication help the virus to further replicate and avoid inflammatory responses and to persist in unnotified progression [22]. This factor is the major force behind the low clearances of viruses in women even after 6–12 months of infections [23]. HPV 16 has been found to inflict long periods of infection. After 12–18 months, the viral clearance occurs systemically by the immune system of the host [22,24]. The infection in the cervical system lasts for 12–18 days following infection with high-risk HPV [22]. In most cases, it clears out spontaneously. However, infection develops when the immune system is compromised and unable to fight against the repercussions of infectious virus loading [25]. This is mostly the case in women with a compromised immune system. In such cases, the virus persists in the form of long-term infection that ultimately progresses to malignant lesions (cervical cancer) [26].

### 3.3. Carcinogenic Cellular Transformation by HPV

HPV is associated with multiple cancer incidences, inducing cervical, pharyngeal, vulvar, vaginal, penile, oral, oropharyngeal, and squamous cell carcinomas, and head and neck cancers [4]. The scientific community has been continuously engaged in exploring the various aspects underlying the mechanism of progression of HPV-associated malignancies [22]. These aspects may include: understanding the latent stages of HPV; the mechanism of development of immunity against reinfection; weak immune responses to HPV infections in terms of the lower expression of inflammatory proteins. Moreover, they are extending studies to look deeper into the persistent and transient infections caused by HPV.

The great majority of HPV infections are resolved within 2 years, while approximately 10% of hosts suffer from persistent HPV infection that leads to lesions and malignancies [22]. The carcinogenic developments are associated with compromised host immune responses as the virus creates genomic deregulations in host cells after multiple cycles of replication, as explained earlier [27]. HPV-16 has been identified as the most promising promoter of HPV-provoked malignancies such as anogenital, oropharyngeal, and, most importantly, cervical cancers [22,28]. The carcinogenicity is promoted by the activation of oncogenic viral proteins E6 and E7 that produce deregulated cellular regulatory networks. These oncogenes lead to a slow host genetic mutation that persists unnoticed for a long period [22]. During this time, the immune system remains unreactive due to the silent nature of viral loading, which is pertinent to the further understanding [28,29].

It is pertinent to mention that mucosal subtypes, such as alpha HPVs, are potential causes of malignancies [6]. Low-risk HPV, which may include HPV 6, 11, 40, 42, 43, 44, 54, 61, 70, 72, and 81, limits infection to benign lesions and rarely progresses to malignant tumors [30]. High-risk HPVs, including HPV 16, 18, 31, 33, 35, 39, 45, 51, 52, 56, 58, 59, and 66, are notified as the primary cause of cervical and other known cancers [30,31]. In both benign and malignant HPV lesions, a high demand for cellular nutrients is extended by HPV, which causes over-expression of hypoxia-inducible factor-1 (HIF-1), and its targeted genes that ultimately induce hypoxia [8,30]. This dysregulation of genes causes induction of angiogenesis, which further persists to produce HPV-induced malignancy by the provision of nutrients for HPV proliferation [32].

The discovery of oncogenic proteins and genes of HPV has allowed scientists to conduct bioinformatics studies and design vaccination and therapeutic protocols against HPV infections. These screening, diagnostics, and treatment-associated studies indicate the potential landscape for understanding and developing treatment options against HPV-related infections and cancers. The latest studies have introduced some vaccination options for the prevention of cervical cancer. These vaccinations also exhibit prophylactic potential against the development of HPV-associated HNSCCs [33]. Studies are also underway to elaborate upon the etiology of beta-HPV for its linkages found with non-melanoma skin cancer [12,34]. In addition to vaccination, the local excision of tumors coupled with typically immune-modulating anti-viral and anti-tumoral drugs is often proposed before adopting the surgical option [35].

Most studies have indicated HPV’s pronounced association with cervical cancer [33]. The scientific community is continuously engaged to develop therapeutics for the treatment of cervical cancer as it is counted among the most commonly occurring cancers caused by HPV [36]. Second in the line of HPV-induced cancers is head and neck squamous cell carcinomas (HNSCCs), which arise in the digestive tract of hosts. HPV-related HNSCCs are different from those that lack HPV intrusion, based on underlying oncogenic regulations, prognosis responses, and available treatment options [37]. Similarly, the concomitant HPV and squamous intra-epithelial lesions (HSILs) have been found in patients suffering from HIV, which also indicates the co-infectious nature of HPV with other viral diseases [23]. Several recent studies have elaborated the clinical reports of patients suffering from primary vulvovaginal cancer and HPV infections and have, thus, established links between HPV infection and vaginal cancer [38,39]. Data from the latest publications explain the associations and incidences of primary oropharyngeal cancers and penile cancer following HPV infection in patients [35]. A formal meta-analysis is deemed important to incorporate a massive research track for these carcinogenic studies. We have tried to cover most of the aspects of HPV-induced cancers (types) and incidences, based upon the available research studies. Study links have also been established with the co-infection of HPV with other viruses in the host body.

Reports show approximately 0.5 million cases of cervical cancer per year [40]. HPV has been indicated as a major cause behind these malignancies by the WHO [40,41]. The standard therapy for cervical cancers in its early stages is the combination of surgery occupied with radio and chemotherapies with an overall 5-year survival rate, while patients with late-stage cancer have a very limited survival time and rate. In most cervical and HNSCCs carcinomas, high-risk HPVs are found to be the risk factor in addition to other cancer predispositions [40]. Specifically in the case of oropharyngeal carcinoma, 70% of cases have been reported as HPV-positive [7,42]. Additionally, several skin cancers have demonstrated beta-HPV-associated carcinoma progression, but the etiology of HPV associated with cutaneous cancer is not clear [43,44]. Similarly, patients with HIV and those receiving immunosuppressive treatment are at high risk of HPV base opportunistic infections [45,46]. Furthermore, approximately 90% of squamous cell carcinomas undergo HPV infection through types 5 and 8 [47]. Moreover, the HPV-beta subtype has been reported to be associated with non-melanoma skin cancer (NMSC), which is a frequent human malignancy worldwide [45,46]. In addition to HPV infection, several other factors play a vital role toward malignancy development, the presence of which heightens persistent HPV infection in hosts [47]. These factors may include: vast exposure to radiations, oxidative stress, UV exposure, and other epigenetic factors such as obesity, stress, and prolonged diseased conditions [43]. To backup these inferences, several studies have demonstrated how HPV 5 and 8 lead to the E6 protein, which initiates UV-B cellular sensitivity and disrupts DNA repair mechanisms in host cells, which subsequently leads to carcinogenesis [47]. A lack of tobacco and alcohol addiction may increase survival rates in these patients [39]. Despite the extreme disease tool, no screening routes have been established yet for accurate early HPV detection. In most of these carcinomas, HPV-positive cases have better prognosis and sensitivity to chemotherapy as compared to HPV-negative tumors [48]. Critical management of such malignancies involves surgical removal, radiotherapy, chemotherapy, and laser treatment options at early stages [47]. A search for less invasive therapies is underway continuously; meanwhile, some immune regulatory agents such as imiquimod and photodynamic therapy have also shown efficiency toward treatment of HPV-induced carcinogenesis [49,50].

To summarize a clear functional manifestation, α- and β-HPV oncologies have been studied. Detailed studies have been conducted on tonsillar, oropharyngeal, and cervical cancers to gain knowledge about HPV-dependent oncogenesis [51]. These studies suggest HPV 16 and 18 play a critical role in these malignancies. In contrast, the relationship between NMSC and β-HPV remains elusive and needs to be further explored [51]. Moreover, HPV has also been detected in the lung, breast, brain, and bladder [52]. However, the data are limited and, therefore, further research studies are needed to establish a strong link with the HPV subtype, oncogenesis, and related treatment options. Some of the important recent studies that represent the carcinogenic occurrences in HPV patients are included in Table 1.

### 3.4. Co-Infection with Other Viruses

A brief overview of important research studies of the past that have elaborated the co-infectious outcomes of HPV with other viruses including HIV, EBV, HSV, and bacteria is included in Table 1. The data pertaining to the number of clinical subjects, method of detection, associated cancers, and clinical outcomes have been included. A brief discussion of these co-infections is below.

#### 3.4.1. Co-Infection of HPV with Human Immunodeficiency Virus (HIV)

Studies have indicated that HPV infection often persists and increases the risk of co-infection with other infectious entities such as viruses (HIV, EBV, HSV, etc.) and bacteria. HPV has been extensively studied as an important risk factor for human malignancies. However, the links and etiological basis for co-infection of HPV with other viral diseases continuously remain in research to develop a proper understanding of the process. Patients specifically affected with HIV have a high risk of anogenital dysplasia owing to HPV persistence [43]. HIV and HPV co-infections are common in HIV (PLWH) patients [53]. Moreover, high-risk entities of HPV often lead to cervix and anogenital cancer by co-infection of HPV and HIV in patients [53]. To prove the underlying cellular mechanisms and develop treatment options, scientists are working on patients with HIV and HPV co-infection [54]. The major cause of co-infection is the immune system vulnerability generated in HIV patients, which provides a gateway for HPV abundance and oncogenic progression in such patients. Though no vaccines are yet approved for full vaccination against HIV, effective therapeutic vaccines are in production for HPV, which could benefit for co-infection mitigation concerns [55]. Moreover, screening and diagnostic procedures are being approved to test against the development of cancerous lesions in infected patients. A brief overview of important research studies of the past that have elaborated the co-infectious outcomes of HPV and HIV is part of Table 1.

#### 3.4.2. Co-Infection of HPV and Epstein–Barr Virus (EBV)

Several studies have elaborated on the co-infection of HPV and Epstein–Barr virus (EBV) in patients that were suffering concomitantly from oral cavity cancers. The role of EBV and similar co-infections has also been found in nasopharyngeal, gastric, and lymph epithelial carcinomas [56]. These studies indicated that all patients did not suffer from dual viral infections, yet many patients exhibited co-infection symptoms from at least two of the subtypes of the discussed viruses. Moreover, studies also indicated the development of a triple viral infection (HPV/EBV/MCPyV) in patients with oral cavity cancer with a high incidence rate of differentiated tumor cells as compared to single virally infected cells [57]. The results helped scientists to infer that co-infection with HPV and EBV can be detected in both tumor and non-tumor cells in different carcinomas [58]. This inference led to the hypothesis that such co-infection can also develop in patients suffering from other types of cancers associated with HPV. Researchers are trying to analyze the cellular phenomena underlying this co-infection to formulate effective therapeutics for future treatment options [56].

#### 3.4.3. Co-Infection of HPV with Herpes Simplex Virus (HSV)

Some recent studies have shown the prevalence of co-infection with herpes simplex virus (HSV) among patients suffering from high-risk human papillomavirus (HR-HPV) [59]. High incidence rates for such co-infections have been observed because of the silent seroprevalence of HSV in approximately 80% of the population worldwide [60]. The major risk factors behind such an infection have been risky sexual behavior and a previous history of HPV and HSV infection [61]. A study reported the co-infection with EBV, HPV, CMV, and HSV-1 in patients suffering from diabetes mellitus type 2 (DM2) [59]. The repercussions of the disease spread were linked to the modification of microbiota components that otherwise control the normal functioning of the human body. A slight modification of such microbiota (consisting of bacteria, viruses, fungi, and protozoa) could lead to co-infection with several viral entities, as elaborated earlier. These kinds of microbiota-associated viral infections are most common in the gastrointestinal, oral cavity, respiratory, and vaginal tracts, where microbiota maintain normal functioning of body systems [62]. This inference also establishes the clinical evidence for the co-infection of HPV with bacteria.

#### 3.4.4. Co-Infections of HPV and Bacteria (Microbiota)

As elaborated earlier, studies have shown the co-infection of HPV with bacterial infection owing to disturbances in the microbiota of patients [62]. Another study has revealed the co-infection of HPV and oral bacteria named *Streptococcus anginosus* that sometimes contribute as an infectious risk factor for the progression of oropharyngeal squamous cell carcinoma (OSCC) [63]. Similarly, a study has established co-infection of HPV with 13 key markers of bacterial vaginosis specifically studied in cervical samples [64]. Such studies have indicated the association of human papillomavirus and bacterial vaginosis with increased risk of high-grade squamous intraepithelial cervical lesions. This bacterial vaginosis and viral co-infections have also inflicted high-grade carcinogenic lesions in patients [65]. Under this scenario, the need is to conduct detailed research analyses and to collect more research data to predict the underlying biological mechanisms in this co-infection to demonstrate causes and deduce therapeutics strategies against it.

**Table 1 vaccines-11-00102-t001:** A brief account of recent studies representing HPV-associated viral and pathogenic co-infections in different cancerous incidences.

Sr.No.	Co-Infection of HPV with Other Pathogens	Number of Samples/Patients	Methods of Detection	Clinical Implications	Cancer Incidence	References
1.	HIV/HPV co-infection	56	Antibody capture and chemiluminescent signal detection.	Atypical squamous cells	Anal squamous cell carcinoma (ASCC), low-grade squamous intraepithelial lesion	[66]
2.	HIV-1, HTLV-1, and other Oncogenic viruses’ co-infection (EBV, HBV, HCV, HDV, and HPV)	---	Systematic analyses	Neurocognitive disorders, Neuroinflammation, Neurodegeneration and cancer	HIV-1-associated cancers and adult T cell leukemia/lymphoma (ATL)	[67]
3.	Co-infection with six pathogenic stis, HIV-1 and HSV 1 and 2	205	Multiplex PCR STD direct flow chip assay and Hybrid Capture-2 assay.	Co-infection with ≥2 pathogens (52.7%)	Cervical cancer	[68]
4.	HIV/HPV co-infection	300	Prospective cohort study.	HPV persistence −46%	Cervical intraepithelial neoplasia Grade 2	[69]
5.	HIV/HPV co-infection	51 articles	Systematic review	Prevalence of HPV (41%), HPV52 (17%), and HPV58 (14%).	Anal and cervical cancer	[70]
6.						
7.	HPV/EBV Co-infection	166	Type-specific PCR/nested-PCR and sequencing	2 cases involved co-infection	Head and neck squamous cell carcinoma	[71]
8.	Co-infection with HPV, EBV and Merkel Cell polyomavirus (MCPYV)	144	Quantitative real-time PCR	Infection with at least two viruses 21.1%	Oral irritation fibroma and Oral squamous cell carcinoma	[57]
9.	HPV/EBV Co-infection	63	Genotyping using SPF10 PCR-DEIA-lipa25 system.	Co-infection (57.9%)EBV and HPV persistent infections	Oropharyngeal cancer (OPC)	[72]
10.	HPV, EBV and Candida albicans co-infection	30	Histological analyses and PCR	EBV (73.3%), HPV (43.3%), and *C. Albicans* (23.3%). Oral leukoplakia with dysplastic changes.	Oral leukoplakia	[73]
11.	HPV/EBV Co-infection	63	--	Epstein–Barr virus (EBV) and oral HPV16/18 persistence	Oral carcinoma	[74]
12.	HPV/EBV Co-infection	90	In situ hybridization with commercial EBER1 and HPV16/18 probes	10% EBV/HPV co-infection	Nasopharyngeal carcinoma	[75]
13.	HPV/EBV Co-infection	84	Microarray data analyses and cell line cultures	EBV and HPV co infection (27.4%) in tumor tissues and (4.8%) in normal tissues	Oral Squamous-Cell Carcinoma	[76]
14.	HPV/EBV Co-infection	108	Real-time PCR	EBV (27.8%) HPV (13%) and Co-infection by EBV and HPV (5.6%)	Oral cancer	[77]
15.	Co-infection of HPV with HSV and (*Chlamydia trachomatis, Trichomonas vaginalis*)	300	Polymerase chain reaction (PCR).	At least 3% different co-infections	Cervical cancer	[78]
16.	EBV, HPV and HSV co-infections	319	Enzyme-linked immunosorbent assay (ELISA) and reverse transcriptase-polymerase chain	(EBV and HPV Co-infection) = (1.6%), (EBV and HSV-2 co-infection) = (14.4%), (HPV and HSV-2 co-infection) = (6.9%) and (EBV, HPV and HSV-2 tri-infection) = (16.6%).	Mono, di and tri- viral infections	[79]
17.	HPV, HSV, and Chlamydia trachomatis co-infection	318	Polymerase Chain Reaction method and Restriction Fragment Length Polymorphism (RFLP)	The HPV prevalence (42%) C. Trachomatis (16%) HSV (3%) and co-infections HPV-C (4%)	Oral squamous cell carcinomas (OSCC)	[80]
18.	HPV/HSV co-infection	137	Multi-site HPV testing and P18 Cohort Study assessments.	HPV and HSV co-infection 95%	Anal cancer	[81]
19.	Infection by Cytomegalovirus, Epstein–Barr virus (EBV), HSV 1 and 2, 6, varicella zoster virus and HPV	18	Real-time polymerase chain reaction	EBV (33.3%) and Human herpesvirus 6 (16.7%)	Adult-onset recurrent respiratory papillomatosis (AORRP)	[82]
20.	Co-infections of high-risk human papillomavirus (hrhpv) and important mycoplasmas including *Mycoplasma hominis*, M. Genitalium, Ureaplasma urealyticum and *U. Parvum*	283	Polymerase chain reaction (PCR) and real-time PCR (rt-PCR)	Hrhpv 12.7% and mycoplasmas 53.7%	Low-Grade Squamous Intraepithelial Cervical Lesions	[83]
21.	HPV co-infection with stis and bacterial vaginosis (BV)		Real-time (RT) PCR assays	BV- and/or STI-positive > 50%	Cervical cancer.	[84]
22.	Mycoplasma hominis (Mh) and Ureaplasma urealyticum (Uu) HPV co-infection	120	Molecular analyses	HPV (83.9%). HPV and Uu co-infection higher in invasive cancer.	Cervical intraepithelial neoplasia and carcinoma	[85]
23.	HPV co-infection with *Ureaplasma* spp., *Mycoplasma* spp., *Chlamydia trachomatis,* and *Neisseria gonorrheae*	44	PCR assays	HPV 75% and 22.73% co-infection with *Ureaplasma* spp. and 9.09% with *Mycoplasma* spp.	Cervical cancer	[86]

### 3.5. Prospective Therapeutics Strategies against HPV-Associated Health Complications

The association of HPV with several human carcinomas necessitates the development of prospective therapeutics to reduce global morbidity and mortality rates linked with HPV. Thus, to share the HPV-based healthcare burden, certain researchers are trying to deliberate a therapeutic vaccine and drug adjuvant against HPV. Therapeutic vaccines could generate humoral and cellular immune responses against HPV infection [27]. For this cause, scientists have formulated some vaccine modalities in the form of live attenuated-vector, protein, and peptide-based sub-unit, dendritic cell, and DNA-based vaccines [87]. Some of these preventive vaccines have been approved against HPV by the FDA. These vaccines are commercially available and designated as bivalent, quadrivalent, and nonavalent vaccines that contain virus-like particles (VLPs). 

These VLP-based proteins are basically made by the self-assembling of L1 capsid proteins. They mostly target HPV-16 and HPV-18, which are regarded as the major cancer-causing subtypes of HPV. One such vaccine has been introduced to the market as monovalent, L1-VLP. This vaccine has shown significant success in preventing the beginning of infection as predicted by post-vaccination follow-ups. However, it lacks the therapeutic effect on pre-existing HPV infections and associated malignancies [88]. The dramatic prophylactic impact of these therapeutic vaccines has presented hope for future vaccination drives for eradication of HPV. However, the associated limits with these vaccines, such as storage complications, costly formulations, and distribution conditions, complicate the process and needs to be dealt with effectively for designing a better vaccination program in the future.

Some other vaccines against HPV have been approved by the FDA and are available in the USA. These may include Gardasil, Gardasil 9, and Cervarix [89]. These vaccines protect against viral subtypes 6, 11, 16, and 18 and provide some cross-protection against HPV types 31, 45, and 52. Currently, there is a lack of information regarding the efficacy of these vaccines against specific cancers and on patients with high-risk HPV exposure [46]. Additionally, these vaccines are limited to high-income countries, while the vaccine acquisition in middle- and low-income countries is limited where the actual disease burden prevails. Thus, there is a need to ensure vaccine availability in high-prevalence regions and to further advance our knowledge and understanding of diagnostic and treatment options for HPV-persistent cancers so that better clinical management is governed [5].

### 3.6. Future Vaccination Focus for HPV-Associated Afflictions

The major focus of future vaccination drives and therapeutics is being kept on ontogenetic targeting. Multi-drug resistance and secondary adverse effects of discovered therapeutics have motivated scientists to employ nanotechnology for combination therapeutic approaches and vaccine design in the future [90]. These nano-based drugs have the ability to increase effectiveness and minimize the number and amount of drug dosages required. Furthermore, the nano-delivery vehicles for the HPV vaccine boosts its efficacy, thus making nano-delivery and drug manufacturing approaches the best possible solution for HPV-associated malignancies and lesions [91].

Another approach for vaccination against HPV is a DNA-based vaccine. DNA vaccines are being designed based on plant-based natural compounds that contain proteins that encode for specific gene sequences with tumor suppressor capabilities. These products are used alone or in the form of combinatorial therapies with some other plant-based drugs that contain anticancer metabolites such as flavonoids, tannins, and triterpenoids [92]. DNA vaccination results in antigen-specific immune responses in subjects and, thus, provides a broader application against HPV and associated cancer therapy. Additionally, DNA-based vaccines are cost-effective, safer, and more stable for production and usage, are easier to manufacture, and can induce more specific immune responses [93]. For these reasons, DNA vaccines are regularly being tested and projected to combinatorial treatments with plant metabolites or other immunomodulatory agents that hold the potential to interfere with carcinogenic cellular pathways. In addition to infection prevention, these drugs help to sensitize the cell for radiotherapy and chemotherapy and provide slight resistance against tumor development [94].

Similarly, plant-based constituents such as flavonoids, tannins, and triterpenoids have been checked for anticancer and cytotoxic activities against different cancerous cell lines such as HeLa cells, A549 (adenocarcinoma), MCF7 (human breast carcinoma), A375 (human skin melanoma), and skin papilloma cell lines [95]. Thus, they can be utilized separately and in combination with other proven anticancer drugs for treatment of HPV-associated malignancies. Plant-based constituents have been concurrently checked for their toxicity profiling, which is minimal against normal cell lines [96]. These plant constituents have also proven their antiproliferative potential in different cell lines and mouse models. Moreover, the immunoregulatory function of some polysaccharides extracted from plants has been studied as a beneficial anticancer activity for having immunoregulatory functions in terms of increased phagocytosis and serum hemolysin levels. These predictions established that plant-based constituents can be effectively utilized for preparing anticancer therapeutics, vaccines, and herbal formulations for treating HPV-associated cancer patients [97].

The need is to keep a continuous focus on the development of a more efficacious vaccine against HPV infection, by utilizing advanced approaches of nanobiotechnology and to have international coverage of educational and awareness programs for preventive and treatment suggestions for the community. These efforts are, however, limited at the initial stages. Scientists are trying to develop a multimodality treatment by combining immunotherapy with plant-derived pharmaceutical agents. These plant-based components and nanotechnology are a new spectrum for scientists to delve into therapeutic options against HPV and associated infections.

## 4. Conclusions

HPV holds a high oncogenic and co-infection potential for several viral and cancerous diseases. This sort of biological manifestation is dangerous for communities. High-income countries are successfully planning, executing, and implementing better research and experiments for designing drugs and conducting vaccination protocols against HPV-associated diseases. However, in the cases of middle-income and low-income countries that are already suffering from significant economic and healthcare burdens, this level of research is not possible. Thus, it is imperative for well-developed nations to ensure approved drugs and vaccination availability in affected regions, which are unable to handle their healthcare burden alone. If ignored, these regions could serve as the center of HPV-associated disease spread throughout the world via globalization. Moreover, it is imperative for the scientific community, medical researchers, educationists, and clinicians to coordinate well and share data and new scientific comprehensions on a wider-scale network. This step will be helpful for the larger cause of scientifically advanced treatment options and public awareness for mitigation and adaptive responses against the HPV-related health burden. It is hoped that in the future, there will be more awareness about HPV disease causes and prevention in the general public and better treatment options to curtail the shadows of the HPV-based healthcare burden at a global level.

## Data Availability

Not applicable.

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
