# Peer review of "Tracking HPV Infection, Associated Cancer Development, and Recent Treatment Efforts—A Comprehensive Review"

_vaccines, 2023, doi:10.3390/vaccines11010102_

Round 1
Reviewer 1 Report
You have tackled a very important and complex study area so congraulations. However, much of your text is very general and without persuasive research outcome details.
Method
“Literature of only English origin has been made part of this study.”
[Please repeat your search in all languages. This may turn up some useful articles with English abstracts then you can email the authors for details, Please repeat your search in international trials registers and Cochrane CENTRAL (> 500,000 RCTs)]
Results
“but in a limited number of cases serious and long-term infection 84 leads to the most critical health-related conditions [8]. Low-risk HPV leads to minor be- 85 nign hyperproliferative lesions, while high-risk HPV mostly leads to malignant lesions”
[This is key to your study. Please provide details of study design, eligibles, participant, numbers, %s and risk of bias].
“HPV- Host interaction initiates with the viral entry into the undifferentiated cells in 104 the squamous epithelium of ectocervix. The process mostly occurs as a result of a high 105 grade mechanical stress or chemical shock that the body may experience [11]. The effected 106 cells (that undergo shock absorption and undergoes viral entry) retains the ability to mul- 107 tiply and differentiate. This property is utilized by viral particles to replicate their genome 108 within the host cells.”
“Basel epithelium, ow- 115 ing to mechanical jerk or chemical causes of entry.”
[please specify in detail what you mean by the unusual terms here of shock and jerk]
“HPV is associated with multiple cancer incidences, inducing cervical, pharyngeal, vulva, 166 vagina, enile, oral, oropharyngeal, squamous cell carcinoma, and head and neck cancers 167 [4]. The scientific community has been continuously engaged to explore the different as- 168 pects underlying mechanism of progression HPV associated malignancies. These aspects 169 may include: understanding the latent stages of HPV, mechanism of development of im- 170 munity against reinfection and weak immune responses to HPV infections in terms of 171 lower expression of inflammatory proteins. Moreover, they are extending studies to look 172 deep into the persistent and transient infections caused by HPV.”
[Again, this is key to your study. Please provide details of study design, eligibles, participant, numbers, %s and risk of bias].
“The discovery of oncogenic proteins and genes of HPV has allowed scientists to con- 195 duct bioinformatics studies and design vaccination and therapeutic protocols against 196 HPV infections. These screening, diagnostics, and treatment associated studies indicate 197 the potential landscape for understanding and developing treatment options against 198 HPV-related infections and cancers. Some vaccination regimens have been introduced in 199 the market for the prevention of cervical cancer which has shown some prophylactic po- 200 tential against development of HPV-associated HNSCCs. Some other studies are estab- 201 lished to understand the etiology of beta-HPV for its linkages found with nonmelanoma 202 skin cancer [13]. Besides vaccination, local excision of tumors is often proposed coupled 203 with typically immune-modulating anti-viral and anti-tumoral drugs before the surgical 204 option [29].”
[this is very general and vague. Please provide details of study design, eligibles, participant, numbers, %s and risk of bias].
“Yearly reports show approximately 0.5 million cases of cervical cancer. HPV has been 220 indicted as a major cause behind these malignancies by WHO [33]. The standard therapy 221 for cervical cancers in its early stages is the combination of surgery occupied with radio 222 and chemotherapies with an overall 5-years survival rate, while patients with late-stage 223 cancer have a very limited survival time and rate. In most cervical and (HNSCCs) carci- 224 nomas, high-risk HPVs are found to be the risk factor besides other cancer predisposi- 225 tions. Specifically in the case of oropharyngeal carcinoma, 70% of cases have reported 226 HPV-positive. Despite the extreme disease tool, no screening routes have been established 227 yet for accurate early HPV detection. In most of these carcinomas HPV positive cases have 228 better prognosis and sensitivity to chemotherapy as compared to HPV-negative tumors 229 [34]. 230 Several skin cancers have demonstrated incidences of beta-HPV-associated carci- 231 noma progression but the etiology of HPV associated with cutaneous cancer is not clear 232 [35]. Patients undergoing AIDS and those receiving immunosuppressive treatment are at 233 high risk of HPV base opportunistic infections [36]. Approximately 90% of squamous cell 234 carcinomas undergo HPV infection through types 5 and 8. Other factors like vast exposure 235 to ultraviolet radiation have been demonstrated as a major factor in these malignancies' 236 Viruses 2022, 14, x FOR PEER REVIEW 6 of 11 development [36]. Some studies have demonstrated how HPV 5 and 8 lead to the E6 pro- 237 tein which initiates UV-B cellular sensitivity and disrupts DNA repair mechanisms in host 238 cells. Lack of tobacco and alcohol addiction may increases survival rates in these patients 239 [37]. Moreover HPV- beta subtype has been reported to be associated with Non-mela- 240 noma skin cancer (NMSC) which is a frequent human malignancy worldwide [38] [39]. 241 Critical management of such malignancies involves surgical removal, radiptherapy, 242 chemotherapy and laser treatment options at early stages. Search for less invasive thera- 243 pies is going on continuously, meanwhile some immune regulatory agents such as 244 imiquimod and photodynamic therapy have also shown efficiency toward treatment [40]. 245 To summarize a clear functional manifestation of α- and β-HPV oncologies have 246 been exhibited by research studies. Detailed studies have been conducted on tonsillar, 247 oropharyngeal, and cervical cancers to gain knowleadge about HPV-dependent oncogen- 248 esis. These studies suggest HPV 16 and 18 plays a critical role in these malignancies. In 249 contrast, the relationship between NMSC and β-HPV remains elusive and needs to be 250 further explored. Moreover HPV have also been detected in lung , breast, brain and blad- 251 der [41] . But the data is limited and therefore further research studies are needed to es- 252 tablish a strong link with the HPV subtype, oncogenesis, and related treatment options”
[Again, a series of very general paragraphs. Please provide details of study design, eligibles, participant, numbers, %s and risk of bias].
[Your entire section on viral coinfections is potentially very important, but is again very general without details of specific studies such as number of patients, methods of detection, period of observation and clinical outcomes]
These are the sections that need intensive rewriting
“3.4. Co-infection with other viruses (HIV, EBV, HSV) 254 3.4.1. Co-infection of HPV with Human Immunodeficiency Virus (HIV) 255 HPV infection often persists and increases the risk of coinfection with other infectious 256 entities like viruses (HIV, EBV, HSV, etc) and bacteria. HPV has been extensively studied 257 as an important risk factor for human malignancies. However, the links and etiological 258 basis for co-infection of HPV with other viral diseases, continuously remain in research 259 annals to develop proper understanding. Patients specifically affected with HIV un- 260 dergo a high risk of anogenital dysplasia owing to HPV persistence [36]. Moreover, high- 261 risk entities of HPV often lead to the cervix and anogenital cancer by co-infection in HIV 262 patients. HIV and HPV co-infections are rather common in HIV (PLWH) patients [42]. To 263 prove the underlying cellular mechanism and develop treatment options, scientists are 264 working on patients with HIV and HPV coinfection [43]. The major cause for the co-infec- 265 tion is the immune system vulnerability generated in HIV patients that provides a gate- 266 way for HPV abundance and oncogenic progression in such patients. Though no vaccines 267 are yet approved for full vaccination against HIV, effective therapeutic vaccines are in 268 production lines for HPV which could benefit for co-infection mitigation concerns [44]. 269 Moreover, screening and diagnostic procedures are being approved to rest against the 270 development of cancerous lesions in infected patients. 271 3.4.2. Co-infection of HPV and Epstein –Barr virus (EBV) 272 Several studies have elaborated on the coinfection of HPV and Epstein –Barr virus (EBV) 273 in patients that were suffering concomitantly from oral cavity cancers. The role of EBV 274 and similar co-infections has also been found in nasopharyngeal, gastric, and lymph epi- 275 thelial carcinomas [45]. These studies indicated that, all patients didn’t suffered from dual 276 viral infections yet major patients exhibited co-infection symptoms from at least two of 277 the subtypes of discussed viruses. Moreover, studies also indicated the development of a 278 triple viral infection (HPV/EBV/MCPyV) in patients with oral cavity cancer with a high 279 incidence rate of differentiated tumor cells as compared to single virally infected cells. The 280 results helped scientists to infer that co-infection with HPV and EBV can be detected in 281 both tumor and non-tumor cells in different carcinomas [46]. Moreover, this inference led 282 to the hypothesis that such co-infection can also develop in patients suffering from other7 of 11 types of cancers associated with HPV. Researchers are trying to analyze the cellular phe- 284 nomena underlying this co-infection to formulate effective therapeutics for future treat- 285 ment options [45]. 286 3.4.3. Co-infection of HPV with Herpes simplex virus (HSV) 287 Some recent studies have shown the prevalence of confection with herpes simplex virus 288 (HSV) among patients suffering from high-risk human papillomavirus (HR-HPV) [47]. 289 High incident rates for such co-infection have been observed. The major risk factors be- 290 hind such infection have been risky sexual behavior and a previous history of HPV infec- 291 tion [48]. A study reported the co-infection with EBV, HPV, CMV and HSV-1 in patients 292 suffering from diabetes mellitus type 2 (DM2) [47]. The repercussions of the disease 293 spread was linked to the modification of microbiota components which otherwise control 294 the normal functioning of human body. A slight modification of such microbiota (consist- 295 ing of bacteria, viruses, fungi and protozoa) could lead to co-infection with several viral 296 entities as elaborated earlier. This kind of micro-biota associated viral infections are most 297 common in gastrointestinal, oral cavity, respiratory and vaginal tracts, where microbiota 298 maintain normal functioning of body systems [49]. 299 3.4.4. Co-infections of HPV and Bacteria (Microbiota) 300 Another study has revealed the coinfections of HPV and oral bacteria named Streptococcus 301 anginosus that sometimes contributes as infectious risk factor for progression of oropha- 302 ryngeal squamous cell carcinoma (OSCC) [50]. Similarly, a study has established co-infec- 303 tion of HPV with 13 key markers of bacterial vaginosis specifically studies in cervical sam- 304 ples [51]. This study indicated the association of human papillomavirus and bacterial 305 vaginosis with increased risk of high-grade squamous intraepithelial cervical lesions. This 306 bacterial vaginosis and viral coinfections are also studied to have inflicted high-grade car- 307 cinogenic lesions in patients [52]. Under this scenario, the need is to conduct detailed re- 308 search analyses and to collect more research data to predict the underlying biological 309 mechanisms in this co-infection to demonstrate causes and deduce therapeutics strategies 310 against it.”
[You suggest an important new research focus on plants. Can you provide specific details please?]
“Another approach for vaccination against HPV is DNA-based vaccination drive. 349 DNA vaccines are being designed based on plant-based natural compounds that contain 350 proteins, which encode for specific gene sequences with tumor suppressor capabilities. 351 These products are used alone or in the form of combinatorial therapies with some other 352 plant metabolites [58]. DNA vaccination results in antigen-specific immune responses in 353 subjects and thus provides a broader application against HPV and associated cancer ther- 354 apy. Additionally, DNA-based vaccines are rather cost-effective, safer, and more stable 355 for production and usage, undergo easier manufacturability, and can induce more specific 356 immune responses [59]. For these reasons, DNA vaccines are excessively being tested and 357 projected to combinatorial treatments with plant metabolites or other immunomodulatory 358 agents that hold the potential to interfere with carcinogenic cellular pathways. Besides 359 infection prevention, these drugs help to sensitize the cell for radiotherapy and chemo- 360 therapy and provides slight resistance against tumor development [60].”
Sentence requiring rewriting
“While some other re- 79 ported non-coding proteins are also part of HPV genomic transcripts, which play role in 80 the virus-host interaction and the infection development via protein-protein interactions”
Typo
Basel epithelium, ow- 115 ing to mechanical jerk or chemical causes of entry. (replace with basal. Basel is a city in Switzerland)
Author Response
You have tackled a very important and complex study area so congratulations. However, much of your text is very general and without persuasive research outcome details.
Response: Thank you for the valuable comments. Manuscript is revised accordingly.
Method
“Literature of only English origin has been made part of this study.”
[Please repeat your search in all languages. This may turn up some useful articles with English abstracts then you can email the authors for details, Please repeat your search in international trials registers and Cochrane CENTRAL (> 500,000 RCTs)]
Response: we have made our article inclusion strategy more inclusive and several other articles have also been made part of this review, as per the instructions.
Results
“but in a limited number of cases serious and long-term infection 84 leads to the most critical health-related conditions [8]. Low-risk HPV leads to minor be- 85 nign hyperproliferative lesions, while high-risk HPV mostly leads to malignant lesions”
[This is key to your study. Please provide details of study design, eligible, participant, numbers, %s and risk of bias].
Response:
Table 1 is added with the details of the study participants, cancer type, methods of detection, co-infection etc.
“HPV- Host interaction initiates with the viral entry into the undifferentiated cells in 104 the squamous epithelium of ectocervix. The process mostly occurs as a result of a high 105 grade mechanical stress or chemical shock that the body may experience [11]. The effected 106 cells (that undergo shock absorption and undergoes viral entry) retains the ability to mul- 107 tiply and differentiate. This property is utilized by viral particles to replicate their genome 108 within the host cells.”
“Basel epithelium, ow- 115 ing to mechanical jerk or chemical causes of entry.”
[please specify in detail what you mean by the unusual terms here of shock and jerk]
Response: by Mechanical stress we mean a stress response that may be experienced in form of an injury to the tissues or bones or extended overloading. While the chemical shock may be experienced in form of an internal disbalanced hormonal and enzymatic profile due to disease state or may be due to external intake of drugs or medicines that effects normal organ regulatory systems of the human body. The same has been explained in the manuscript which has been traced down with proper referencing in the revised file.
“HPV is associated with multiple cancer incidences, inducing cervical, pharyngeal, vulva, 166 vagina, enile, oral, oropharyngeal, squamous cell carcinoma, and head and neck cancers 167 [4]. The scientific community has been continuously engaged to explore the different as- 168 pects underlying mechanism of progression HPV associated malignancies. These aspects 169 may include: understanding the latent stages of HPV, mechanism of development of im- 170 munity against reinfection and weak immune responses to HPV infections in terms of 171 lower expression of inflammatory proteins. Moreover, they are extending studies to look 172 deep into the persistent and transient infections caused by HPV.”
[Again, this is key to your study. Please provide details of study design, eligible, participant, numbers, %s and risk of bias].
response : We have included a detailed table to include the relevant details about specific study designs, eligible participants and other important details for readers to trace down and analyses the research outcomes.
“The discovery of oncogenic proteins and genes of HPV has allowed scientists to con- 195 duct bioinformatics studies and design vaccination and therapeutic protocols against 196 HPV infections. These screening, diagnostics, and treatment associated studies indicate 197 the potential landscape for understanding and developing treatment options against 198 HPV-related infections and cancers. Some vaccination regimens have been introduced in 199 the market for the prevention of cervical cancer which has shown some prophylactic po- 200 tential against development of HPV-associated HNSCCs. Some other studies are estab- 201 lished to understand the etiology of beta-HPV for its linkages found with nonmelanoma 202 skin cancer [13]. Besides vaccination, local excision of tumors is often proposed coupled 203 with typically immune-modulating anti-viral and anti-tumoral drugs before the surgical 204 option [29].”
[this is very general and vague. Please provide details of study design, eligible, participant, numbers, %s and risk of bias].
Response: We have modified the manuscript for the said changes. Proper referencing have been added to back the result inference and table has been added to be more elaborative in our research outcomes.
“Yearly reports show approximately 0.5 million cases of cervical cancer. HPV has been 220 indicted as a major cause behind these malignancies by WHO [33]. The standard therapy 221 for cervical cancers in its early stages is the combination of surgery occupied with radio 222 and chemotherapies with an overall 5-years survival rate, while patients with late-stage 223 cancer have a very limited survival time and rate. In most cervical and (HNSCCs) carci- 224 nomas, high-risk HPVs are found to be the risk factor besides other cancer predisposi- 225 tions. Specifically in the case of oropharyngeal carcinoma, 70% of cases have reported 226 HPV-positive. Despite the extreme disease tool, no screening routes have been established 227 yet for accurate early HPV detection. In most of these carcinomas HPV positive cases have 228 better prognosis and sensitivity to chemotherapy as compared to HPV-negative tumors 229 [34]. 230 Several skin cancers have demonstrated incidences of beta-HPV-associated carci- 231 noma progression but the etiology of HPV associated with cutaneous cancer is not clear 232 [35]. Patients undergoing AIDS and those receiving immunosuppressive treatment are at 233 high risk of HPV base opportunistic infections [36]. Approximately 90% of squamous cell 234 carcinomas undergo HPV infection through types 5 and 8. Other factors like vast exposure 235 to ultraviolet radiation have been demonstrated as a major factor in these malignancies' 236 Viruses 2022, 14, x FOR PEER REVIEW 6 of 11 development [36]. Some studies have demonstrated how HPV 5 and 8 lead to the E6 pro- 237 tein which initiates UV-B cellular sensitivity and disrupts DNA repair mechanisms in host 238 cells. Lack of tobacco and alcohol addiction may increases survival rates in these patients 239 [37]. Moreover HPV- beta subtype has been reported to be associated with Non-mela- 240 noma skin cancer (NMSC) which is a frequent human malignancy worldwide [38] [39]. 241 Critical management of such malignancies involves surgical removal, radiotherapy, 242 chemotherapy and laser treatment options at early stages. Search for less invasive thera- 243 pies is going on continuously, meanwhile some immune regulatory agents such as 244 imiquimod and photodynamic therapy have also shown efficiency toward treatment [40]. 245 To summarize a clear functional manifestation of α- and β-HPV oncologies have 246 been exhibited by research studies. Detailed studies have been conducted on tonsillar, 247 oropharyngeal, and cervical cancers to gain knowledge about HPV-dependent oncogene- 248 esis. These studies suggest HPV 16 and 18 plays a critical role in these malignancies. In 249 contrast, the relationship between NMSC and β-HPV remains elusive and needs to be 250 further explored. Moreover HPV have also been detected in lung , breast, brain and blad- 251 der [41] . But the data is limited and therefore further research studies are needed to es- 252 tablish a strong link with the HPV subtype, oncogenesis, and related treatment options”
[Again, a series of very general paragraphs. Please provide details of study design, eligible, participant, numbers, %s and risk of bias].
Response: a detailed table has been outline that incorporates queries such as co-infection of HPV with other viruses and pathogens, number of participants, methods of detection, clinical implications, cancer incidences and references to trace down the studies. Moreover specific references have been added with each and every result inference made in paragraph section so that they can be traced by the readers for a detailed account.
[Your entire section on viral coinfections is potentially very important, but is again very general without details of specific studies such as number of patients, methods of detection, period of observation and clinical outcomes]
These are the sections that need intensive rewriting
“3.4. Co-infection with other viruses (HIV, EBV, HSV) 254 3.4.1. Co-infection of HPV with Human Immunodeficiency Virus (HIV) 255 HPV infection often persists and increases the risk of coinfection with other infectious 256 entities like viruses (HIV, EBV, HSV, etc) and bacteria. HPV has been extensively studied 257 as an important risk factor for human malignancies. However, the links and etiological 258 basis for co-infection of HPV with other viral diseases, continuously remain in research 259 annals to develop proper understanding. Patients specifically affected with HIV un- 260 dergo a high risk of anogenital dysplasia owing to HPV persistence [36]. Moreover, high- 261 risk entities of HPV often lead to the cervix and anogenital cancer by co-infection in HIV 262 patients. HIV and HPV co-infections are rather common in HIV (PLWH) patients [42]. To 263 prove the underlying cellular mechanism and develop treatment options, scientists are 264 working on patients with HIV and HPV coinfection [43]. The major cause for the co-infec- 265 tion is the immune system vulnerability generated in HIV patients that provides a gate- 266 way for HPV abundance and oncogenic progression in such patients. Though no vaccines 267 are yet approved for full vaccination against HIV, effective therapeutic vaccines are in 268 production lines for HPV which could benefit for co-infection mitigation concerns [44]. 269 Moreover, screening and diagnostic procedures are being approved to rest against the 270 development of cancerous lesions in infected patients. 271 3.4.2. Co-infection of HPV and Epstein –Barr virus (EBV) 272 Several studies have elaborated on the coinfection of HPV and Epstein –Barr virus (EBV) 273 in patients that were suffering concomitantly from oral cavity cancers. The role of EBV 274 and similar co-infections has also been found in nasopharyngeal, gastric, and lymph epi- 275 thelial carcinomas [45]. These studies indicated that, all patients didn’t suffered from dual 276 viral infections yet major patients exhibited co-infection symptoms from at least two of 277 the subtypes of discussed viruses. Moreover, studies also indicated the development of a 278 triple viral infection (HPV/EBV/MCPyV) in patients with oral cavity cancer with a high 279 incidence rate of differentiated tumor cells as compared to single virally infected cells. The 280 results helped scientists to infer that co-infection with HPV and EBV can be detected in 281 both tumor and non-tumor cells in different carcinomas [46]. Moreover, this inference led 282 to the hypothesis that such co-infection can also develop in patients suffering from other7 of 11 types of cancers associated with HPV. Researchers are trying to analyze the cellular phe- 284 nomena underlying this co-infection to formulate effective therapeutics for future treat- 285 ment options [45]. 286 3.4.3. Co-infection of HPV with Herpes simplex virus (HSV) 287 Some recent studies have shown the prevalence of confection with herpes simplex virus 288 (HSV) among patients suffering from high-risk human papillomavirus (HR-HPV) [47]. 289 High incident rates for such co-infection have been observed. The major risk factors be- 290 hind such infection have been risky sexual behavior and a previous history of HPV infec- 291 tion [48]. A study reported the co-infection with EBV, HPV, CMV and HSV-1 in patients 292 suffering from diabetes mellitus type 2 (DM2) [47]. The repercussions of the disease 293 spread was linked to the modification of microbiota components which otherwise control 294 the normal functioning of human body. A slight modification of such microbiota (consist- 295 ing of bacteria, viruses, fungi and protozoa) could lead to co-infection with several viral 296 entities as elaborated earlier. This kind of micro-biota associated viral infections are most 297 common in gastrointestinal, oral cavity, respiratory and vaginal tracts, where microbiota 298 maintain normal functioning of body systems [49]. 299 3.4.4. Co-infections of HPV and Bacteria (Microbiota) 300 Another study has revealed the coinfections of HPV and oral bacteria named Streptococcus 301 anginosus that sometimes contributes as infectious risk factor for progression of oropha- 302 ryngeal squamous cell carcinoma (OSCC) [50]. Similarly, a study has established co-infec- 303 tion of HPV with 13 key markers of bacterial vaginosis specifically studies in cervical sam- 304 ples [51]. This study indicated the association of human papillomavirus and bacterial 305 vaginosis with increased risk of high-grade squamous intraepithelial cervical lesions. This 306 bacterial vaginosis and viral coinfections are also studied to have inflicted high-grade car- 307 cinogenic lesions in patients [52]. Under this scenario, the need is to conduct detailed re- 308 search analyses and to collect more research data to predict the underlying biological 309 mechanisms in this co-infection to demonstrate causes and deduce therapeutics strategies 310 against it.”
[You suggest an important new research focus on plants. Can you provide specific details please?]
“Another approach for vaccination against HPV is DNA-based vaccination drive. 349 DNA vaccines are being designed based on plant-based natural compounds that contain 350 proteins, which encode for specific gene sequences with tumor suppressor capabilities. 351 These products are used alone or in the form of combinatorial therapies with some other 352 plant metabolites [58]. DNA vaccination results in antigen-specific immune responses in 353 subjects and thus provides a broader application against HPV and associated cancer ther- 354 apy. Additionally, DNA-based vaccines are rather cost-effective, safer, and more stable 355 for production and usage, undergo easier manufacturability, and can induce more specific 356 immune responses [59]. For these reasons, DNA vaccines are excessively being tested and 357 projected to combinatorial treatments with plant metabolites or other immunomodulatory 358 agents that hold the potential to interfere with carcinogenic cellular pathways. Besides 359 infection prevention, these drugs help to sensitize the cell for radiotherapy and chemo- 360 therapy and provides slight resistance against tumor development [60].”
Response: Manuscript is edited by a professional language editor from Scotland and English language of paper is much improved. A brief paragraph in result section with a proper reference has been added to elaborate the new concept more profoundly.
Sentence requiring rewriting
“While some other re- 79 ported non-coding proteins are also part of HPV genomic transcripts, which play role in 80 the virus-host interaction and the infection development via protein-protein interactions”
Response: sentence has been restructured to remove the error and confusion
Typo
Basel epithelium, ow- 115 ing to mechanical jerk or chemical causes of entry. (Replace with basal. Basel is a city in Switzerland)
Response: Typo removed and traced
Reviewer 2 Report
The present review reports on studies conducted to understand the HPV virus, infection, associated malignancies and treatment options. The authors reviewed the status of HPV infection worldwide and current areas of research with respect to new therapies and treatment options.
The data presented and literature review, however has multiple limitations and lacunae;
1. The English language used in the review is not at par with scientific standards and needs major revision with respect to grammar and sentence formation.
2. Multiple words are written incorrectly. For eg. HPV subtypes are denoted as categories in line number 35 and 37, strains are written as stains in line number 39.
3. Line number 48 – The data presented is not clear. It does not signify whether data represents worldwide cases or just cases from a particular country.
4. Line number 105-106 – The statement that viral entry occurs due to mechanical stress and chemical shock is not backed up with relevant scientific data? How is it possible that a virus entry into host cells is caused by physical and chemical causes?
5. Line number 115, 128, 144 – Kindly replace basel with basal, outer cellular space as extracellular, viral loads as virus.
6. Line number 323 – A monovalent vaccine is shown to impart immunity against 9 subtypes, how is it possible? Kindly check.
7. Line number 352 – What other plant metabolites mean?
The review article presented does not include a thorough in-depth literature search and review. The language and sentence formation are not proper. Some of the data is not presented correctly. There are multiple repetitions of statements showing an incomplete understanding of the research done. Authors are requested to perform an in-depth study of literature pertaining to topic and add more studies in the review article.
Author Response
The present review reports on studies conducted to understand the HPV virus, infection, associated malignancies and treatment options. The authors reviewed the status of HPV infection worldwide and current areas of research with respect to new therapies and treatment options.
The data presented and literature review, however has multiple limitations and lacunae;
- The English language used in the review is not at par with scientific standards and needs major revision with respect to grammar and sentence formation.
Response: the whole manuscript has been revised for typos, grammar and spelling errors, and removed (if any), moreover it has been now reviewed by an English native professional language editor from Scotland.
- Multiple words are written incorrectly. For e.g. HPV subtypes are denoted as categories in line number 35 and 37, strains are written as stains in line number 39.
Response: pointed mistakes have been removed and modified in manuscript
- Line number 48 – The data presented is not clear. It does not signify whether data represents worldwide cases or just cases from a particular country.
Response: clarification has been made in the revised manuscript.
- Line number 105-106 – The statement that viral entry occurs due to mechanical stress and chemical shock is not backed up with relevant scientific data? How is it possible that a virus entry into host cells is caused by physical and chemical causes?
Response: brief explanation has been added with reference to properly elaborate the statement.
- Line number 115, 128, 144 – Kindly replace basel with basal, outer cellular space as extracellular, viral loads as virus.
Response: changes have been made as suggested
- Line number 323 – A monovalent vaccine is shown to impart immunity against 9 subtypes, how is it possible? Kindly check.
Response: there was a typo error of word “nine” we have made the changes and elaborated the line a little bit to be mre clear.
- Line number 352 – What other plant metabolites mean?
Response: it means the anti-cancerous metabolites in plants (such as flavonoids, tannins, triterpenoids and others) that are used in the form of plant based drugs in combination to approved therapeutic options. A brief paragraph has been added to be more elaborative in manuscript
The review article presented does not include a thorough in-depth literature search and review. The language and sentence formation are not proper. Some of the data is not presented correctly. There are multiple repetitions of statements showing an incomplete understanding of the research done. Authors are requested to perform an in-depth study of literature pertaining to topic and add more studies in the review article.
Response: Thank you for the valuable suggestions, we appreciate the concern. We have tried to inculcate data from more relevant studies and have included proper references with the included data so that confusion can be avoided, and data can be easily traced back from the reference published work. Moreover, the article has again reviewed for language editing and mistakes removal so that it become publishable under the platform of journal Vaccine.
Round 2
Reviewer 1 Report
Thanks to the authors for their work in correcting problems in this manuscript reviewing complex interactions between viruses.
Reviewer 2 Report
Thank you for addressing the raised points.